# Biomimetic Mn(III) porphyrin-catalyzed aromaticity-breaking epoxidation of electron-deficient naphthalene

Hongli Wu[1,3], Jialun Xu[1,3], Junjie Tai[1,3], Jingkun Gao[1], Yunfei Ge[1], Guijie Li [ID][2] ✉, Yuan-Bin She[2] ✉ & Yun-Fang Yang [ID][1] ✉

Selective catalytic aromaticity-breaking epoxidation of arenes remains an ongoing research challenge, attributed to the stability of aromatic systems, the occurrence of over-oxidation, and the instability of epoxides. Epoxides, recognized as short-lived and highly reactive intermediates, are prone to undergo the "NIH-shift" rearrangement to yield the hydroxylated products. Herein, we report the synthesis and characterization of several electron-deficient naphthalene epoxides that disrupt the aromaticity of naphthalene. Their formation is highly dependent on the substrate, especially on the specific substitution pattern. This achievement is enabled by using mild biomimetic manganese porphyrins as catalysts and iodosylbenzene as an oxidant. The strategic employment of electron-withdrawing substitution on naphthalenes enables a chemoselective switch from hydroxylation to epoxidation. Experimental studies and theoretical calculations elucidate the possible underlying mechanisms of epoxidation and hydroxylation: epoxidation is likely mediated by a radical mechanism, whereas hydroxylation predominantly follows a zwitterionic pathway.

Oxidative functionalization of hydrocarbons represents a cornerstone in biological and chemical transformations, playing a vital role in the synthesis of oxygen-containing organic molecules[1–10]. Aromatic systems are inherently thermodynamically and kinetically inert, and the occurrence of over-oxidation during oxidation further complicates the process, rendering selective oxyfunctionalization a challenging yet significant area of research[11–13]. Despite significant advancements in biocatalysis and biomimetic hydroxylation[14–16], the epoxidation reaction remains challenging, primarily due to over-oxidation, aromaticity-breaking, and instability of epoxides. The epoxides are short-lived and highly reactive intermediates in reactions, prone to undergo the "NIH-shift"[17–19] rearrangement to yield the corresponding hydroxylated products (Fig. 1a). This propensity further exacerbates the challenges associated with the synthesis of epoxides, which is particularly

regrettable given their potential as valuable building blocks in chemical transformations and drug metabolism[20–25]. Over the past decade, only a few cases of epoxidation of naphthalene, anthracene, tetracene and acridine have been reported[26–28]. These reactions predominantly result in the formation of diepoxides or peroxides rather than desired monoepoxides. Sarlah et al. have recently reported a general arenophile-based strategy for the oxidative dearomatization of benzenes and pyridines, primarily yields electron-rich arene monoepoxides. (Fig. 1c)[29,30]. However, this strategy, which requires a two-step process and involves the use of N-methyl-1,2,4-triazoline-3,5-dione (MTAD) in combination with potentially explosive peroxyacetic acid. The work of Hollmann group demonstrates that naphthalene epoxide can be directly generated by fungal peroxygenases[31]. Unfortunately, the monoepoxidation of electron-deficient naphthalenes remains

[1]State Key Laboratory of Advanced Separation Membrane Materials, College of Chemical Engineering, Zhejiang University of Technology, Hangzhou, Zhejiang, China. [2]State Key Laboratory of Green Chemical Synthesis and Conversion, College of Chemical Engineering, Zhejiang University of Technology, Hangzhou, Zhejiang, China. [3]These authors contributed equally: Hongli Wu, Jialun Xu, Junjie Tai. ✉e-mail: guijieli@zjut.edu.cn; sheyb@zjut.edu.cn; yangyf@zjut.edu.cn

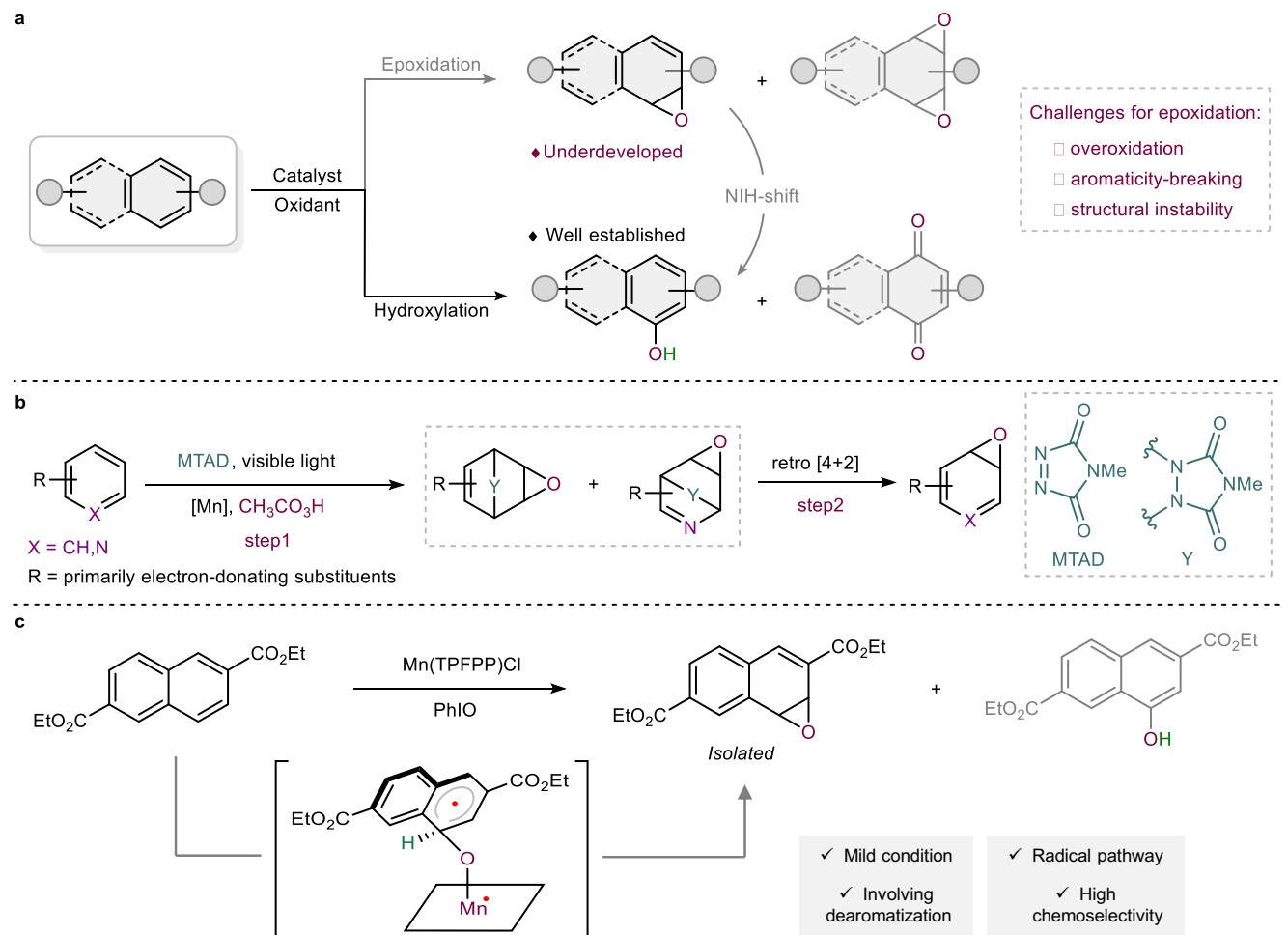

**Fig. 1 | The development of oxidative functionalization of arenes. a** Oxidative functionalization of arenes. **b** Mn-catalyzed two-step epoxidation of benzenes and pyridines. **c** This work: biomimetic catalysis enabled one-step epoxidation of electron-deficient naphthalene. TPFPP: 5,10,15,20-tetrakis(pentafluorophenyl) porphyrin.

largely unexplored. Consequently, the development of more straightforward, mild, and highly selective catalytic epoxidation methods for electron-deficient arenes is in high demand.

Biomimetic metalloporphyrins have been pivotal in oxidative functionalization due to their facile structural modification and low catalyst loading[32–34]. We previously employed the highly fluorinated manganese porphyrin Mn(TPFPP)Cl as a catalyst for the oxidation of electron-rich 2,7-dimethoxynaphthalene, with iodosylbenzene (PhIO) as the oxidant. This process yielded hydroxylated products, which were then further oxidized to the observed quinones[35]. Encouragingly, our present work demonstrates that diethyl-2,6-naphthalenedicarboxylate, featuring a distinct substitution pattern undergoes epoxidation under mild catalytic conditions, enabling the dearomative synthesis of the corresponding epoxide with high chemoselectivity (Fig. 1c). By adjusting the substituents, we accomplished a selective shift from hydroxylation to epoxidation. Notably, while revising our manuscript, Costas and co-workers reported a breakthrough manganese-catalyzed enantioselective diepoxidation of a broad range of arenes[36]. Their success arises from the synergistic combination of an electron-rich, sterically demanding manganese complex and amino acid co-ligands containing a *tert*-butyl-leucine moiety. In contrast, our study employs a highly fluorinated Mn(TPFPP)Cl catalyst operating without any co-catalyst, enabling selective mono- and diepoxidation of electron-deficient naphthalenes. It is particularly intriguing that these distinct catalytic systems afford analogous epoxidation products. Moreover, our work complements this recent advance by providing detailed mechanistic insights into the competition between

epoxidation and hydroxylation pathways, thereby elucidating the origin of substituent-controlled selectivity reversal.

## Results
### Experimental studies

The initial attempts have unveiled the potential of electron-deficient diethyl-2,6-naphthalenedicarboxylate **1a** in epoxidation under the catalysis of Mn(TPFPP)Cl and with PhIO as the oxidant. Subsequently, the influence of oxidant and catalyst loadings, reaction time and temperature on the reactivity and selectivity has been comprehensively evaluated. The representative outcomes are presented in Table 1, while the others are provided in the Supporting Information (SI) (Tables S1–S4). To our delight, epoxide **3a** was formed with no hydroxylation product **5a'** observed (Table 1, entry 1), and the structure of this epoxide has been confirmed by X-ray diffraction (See Table S5 for details). It was observed that the amounts of oxidant PhIO and catalyst Mn(TPFPP)Cl significantly influence the reaction activity and the selectivity (Table 1, entries 2–4 and 7–10). The optimal amounts for PhIO and Mn(TPFPP)Cl are 1.6 equiv and 1.0 mol%, respectively. Further observations reveal that extending the reaction time leads to a decrease in yield of **3a** and an increase for **5a'** (Table 1, entries 5–7). The yield of **3a** is maximized when the reaction duration is controlled at one hour. This epoxide is inherently stable, and its rearrangement is catalytically driven (see SI Fig. S1 for details). Additionally, an assessment of the effect of temperature on the reaction shows that at 40 °C, the conversion yield is the highest, reaching 65% (Table 1, entry 11). However, when the temperature is raised to 80 °C, the selectivity for epoxide formation is optimal (Table 1, entry 9).

**Table. 1 | Reaction optimization**

CCDC 2404381

| Entry | PhIO (equiv.) | Reaction time (h) | Mn(TPFPP)Cl (mol%) | Temperature (°C) | Conversion (%) | Selectivity (%)[a] | |
|---|---|---|---|---|---|---|---|
| | | | | | | 3a | 5a' |
| 1 | 1.0 | 2 | 1.0 | 80 | 54 | 30 | 0.0 |
| 2 | 1.4 | 2 | 1.0 | 80 | 46 | 74 | 2.2 |
| 3 | 1.6 | 2 | 1.0 | 80 | 49 | 98 | 0.0 |
| 4 | 1.8 | 2 | 1.0 | 80 | 36 | 83 | 0.0 |
| 5 | 1.6 | 8 | 1.0 | 80 | 52 | 14 | 34 |
| 6 | 1.6 | 4 | 1.0 | 80 | 49 | 68 | 1.7 |
| 7 | 1.6 | 1 | 1.0 | 80 | 45 | 99 | 0.0 |
| 8 | 1.6 | 1 | 1.5 | 80 | 47 | 90 | 0.0 |
| 9 | 1.6 | 1 | 2.0 | 80 | 42 | 95 | 0.0 |
| 10 | 1.6 | 1 | 3.0 | 80 | 50 | 76 | 0.0 |
| 11 | 1.6 | 1 | 1.0 | 40 | 65 | 78 | 0.0 |
| 12 | 1.6 | 1 | 1.0 | 50 | 52 | 91 | 0.0 |
| 13 | 1.6 | 1 | 1.0 | 60 | 57 | 84 | 0.0 |

Unless otherwise noted, the reactions were performed in the presence of 1a (0.184 mmol) and $CH_3CN$ (2.0 mL). The yields were determined by $^1$H NMR analyses of the crude products using $CH_2Br_2$ as the internal standard. [a]The yields were based on the conversion yields.

Considering both conversion yield and selectivity, the optimal reaction conditions are 1.6 equiv of PhIO, 1.0 mol% Mn(TPFPP)Cl, 50 °C, and 1 h (Table 1, entry 12). Notably, arene oxidation is inherently complex and often generates a variety of side products beyond the reported epoxides and phenols—including dihydroxylated compounds, diepoxides, and quinones. We made efforts to isolate and identify these possible by-products. However, due to the complexity of the reaction mixtures, the presence of isomeric species, and their closely related polarities, effective separation and characterization were not achievable under our current conditions. We believe that these uncharacterized by-products are responsible for the observed mass imbalance.

With optimal conditions established, we systematically investigated the reactivity of various naphthalene derivatives, focusing on both substitution pattern and electronic effects (Fig. 2). Our prior work had already indicated that electron-rich naphthalenes preferentially yield quinone products under similar conditions[35]. Consistent with this, when simple naphthalene was subjected to the current reaction conditions, quinones were obtained as the major products (see Fig. S2 for details), indicating that electron-donating gruops disfavor epoxide formation. We therefore turned our attention to electron-deficient substrates (Fig. 2). Among di-substituted naphthalenes, 2,6- or 2,7-diester-substituted derivatives afforded mono-epoxides 3a, 3b and 3c in moderate yields. In contrast, methyl 6-bromo-2-naphthoate afforded 3d in low yields. Cyano-substituted derivatives such as 2,3- and 1,4-cyano-substituted naphthalenes predominantly formed diepoxides (4e and 4 f), while methyl 5-bromo-2-naphthoate yielded a mixture of epoxide 3g and quinone 5g. For mono-substituted naphthalenes, only sulfonate-substituted substrates (-SO$_3$Ph or -SO$_3$Me derivatives) produced mono-epoxides (3k, 3l), along with diepoxides (4k, 4l). Other electron-withdrawing groups such as -CO$_2$Me, -CN, and NO$_2$ primarily gave diepoxides or mixtures of diepoxides and quinones (4h/5h, 4i/5i, 4j/5j, 4m). Mono-substituted quinoline derivatives bearing -CO$_2$Me or

-CN mainly afforded dioxetanes 4n and 4o. The X-ray crystal structures of 4e and 4j confirmed the anti-configuration of the diepoxides (See Tables S6, 7 for details). Notably, no epoxides were detected for naphthalene-2,6-dicarbonitrile 1p, 2-(trifluoromethyl) naphthalene 1q, or 1-(trifluoromethyl) naphthalene 1r. Experimental results demonstrate that the type and substitution pattern of substituents on the naphthalene ring effectively modulate the selectivity between mono- and diepoxides formation. Substrates 1a–1d and 1g, bearing electron-withdrawing groups at the 2,5-, 2,6-, and 2,7-positions, afforded monoepoxides exclusively without diepoxidation, a behavior ascribed to the combined effects of enhanced electrophilicity and sterically constrained environment at the substituted ring. In contrast, substrates 1e, 1f, 1h–1j, and 1m–1o underwent selective diepoxidation on the unsubstituted ring, highlighting the influence of steric accessibility. Interestingly, the sulfonate-substituted substrates 1k and 1l yielded mixtures of mono- and diepoxides: monoepoxidation occurred preferentially on the substituted ring, likely directed by the sulfonate group, while diepoxidation the unsubstituted ring due to steric accessibility.

## Mechanistic consideration

Despite extensive studies on the diverse and intricate mechanisms of enzymatic hydroxylation of arenes, the debate over the competitive pathways between hydroxylation and epoxidation persists[37–42]. It is widely accepted that aromatic epoxide is an obligatory intermediate for the formation of hydroxylated product[37]. These studies indicate that the Mn-oxo porphyrin intermediate **B** attacks the π-system of **A** through two distinct oxygen atom attack mechanisms: radical and electrophilic pathways. This results in the formation of radical intermediate **C** and zwitterionic intermediate **D**, respectively (Fig. 3a). It is proposed that both intermediate **C** and **D** can undergo a ring closure process (path a) to form the epoxide-coordinated intermediate **E**.

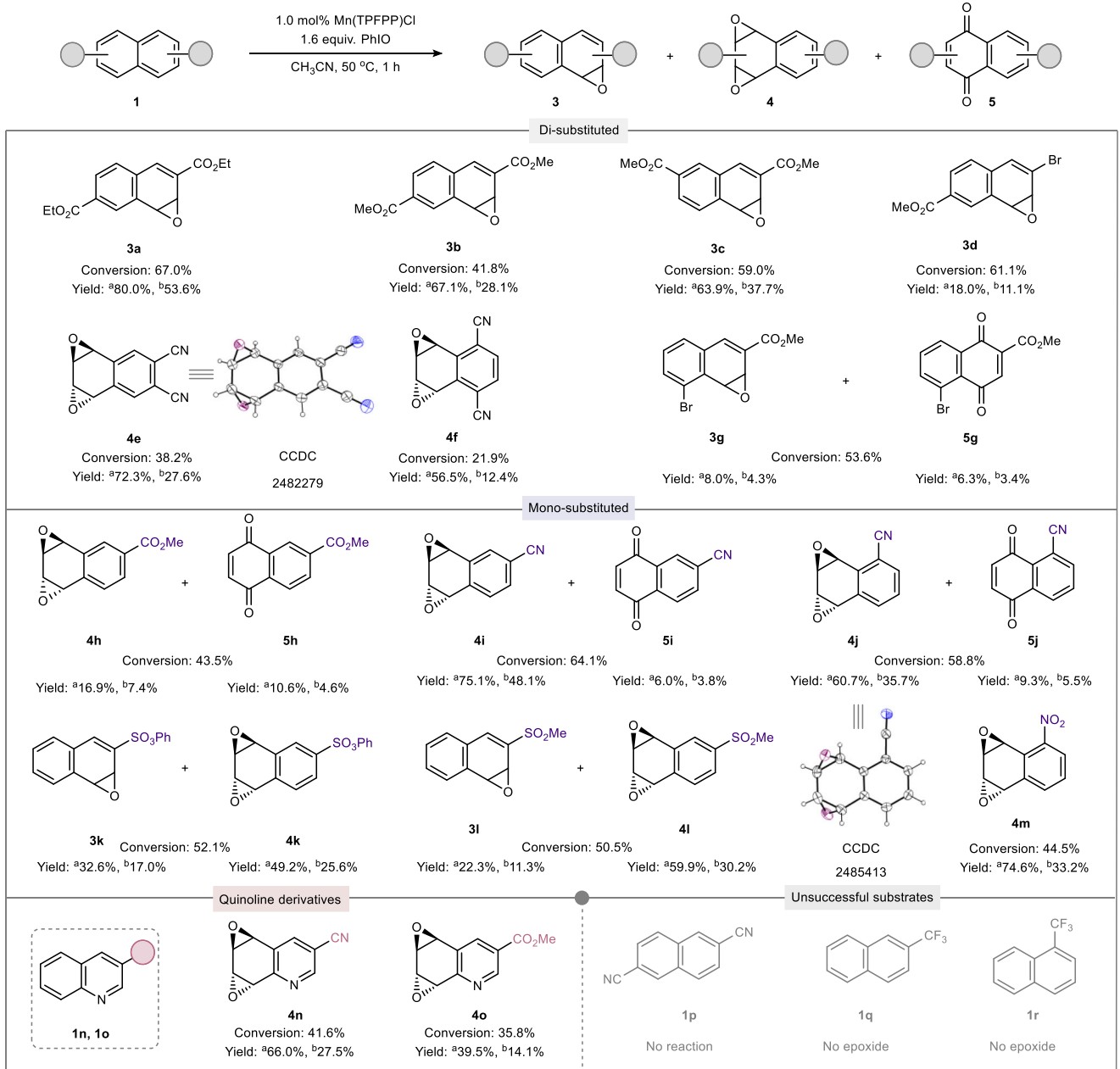

**Fig. 2 | Electron-deficient arenes scope.** [a]The yields were based on the conversion yields. [b]The yields were based on isolated yields.

Alternatively, they can undergo hydroxylation via either an "NIH shift" (path b) or a "proton shuttle" mechanism (path c). The NIH shift involves the intramolecular migration of a hydrogen atom (highlighted in green) to an adjacent carbon atom, yielding the ketone intermediate **F**, which subsequently undergoes keto-enol tautomerization. The proton shuttle mechanism involves a proton transfer mediated by the porphyrin ligand, whereby the proton first migrates to a nitrogen atom on the porphyrin ring to form the N-protonated intermediate **G**, and is then shuttled to the oxygen atom of the Mn-oxo species. After obtaining the epoxide **H**, it may further undergo the NIH shift to form the hydroxylated **I** (path d). It should be noted that path a is defined as either the radical or zwitterionic form, depending on whether it originates from **C** or **D**. Similarly, path b and c also exist in both radical and zwitterionic forms.

The experimental results indicate that the chemoselectivity between hydroxylation and epoxidation can be reversed through the modification of substituents (Fig. 3b). It should be noted that the 2,6-

and 2,7-substitution patterns of **1a** and **1s** have no significant influence on the chemoselectivity (see Fig. S3 for details). Despite the successful isolation of the epoxide without rearrangement to naphthol, the underlying mechanism of its formation and the origin of the reversal of chemoselectivity are yet to be elucidated. To gain more insights into the mechanism of this reaction, a series of experiments and DFT calculations were conducted. First, we monitored the reaction of 2,7-dimethoxynaphthalene **1s** over time. Thin-layer chromatography and mass spectrometry showed that quinone product **5s** formed within 1–10 minutes, with no detectable intermediate spots between **1s** and **5s** including the epoxide **3s** (see Fig. S4 for details). After extended monitoring (10–60 min), no trace of **3s** was detected. (Fig. 3b, left). Additionally, we calculated the relative thermodynamic stability of the epoxide **3** and hydroxylation product **5'** (Fig. 3b, right). For substrate **1a** and **1s**, the hydroxylation products **5a'** and **5s'** were more stable than the corresponding epoxide **3a** and **3s** by 37.1 and 36.0 kcal/mol, respectively. This suggests that both electron-rich and electron-poor

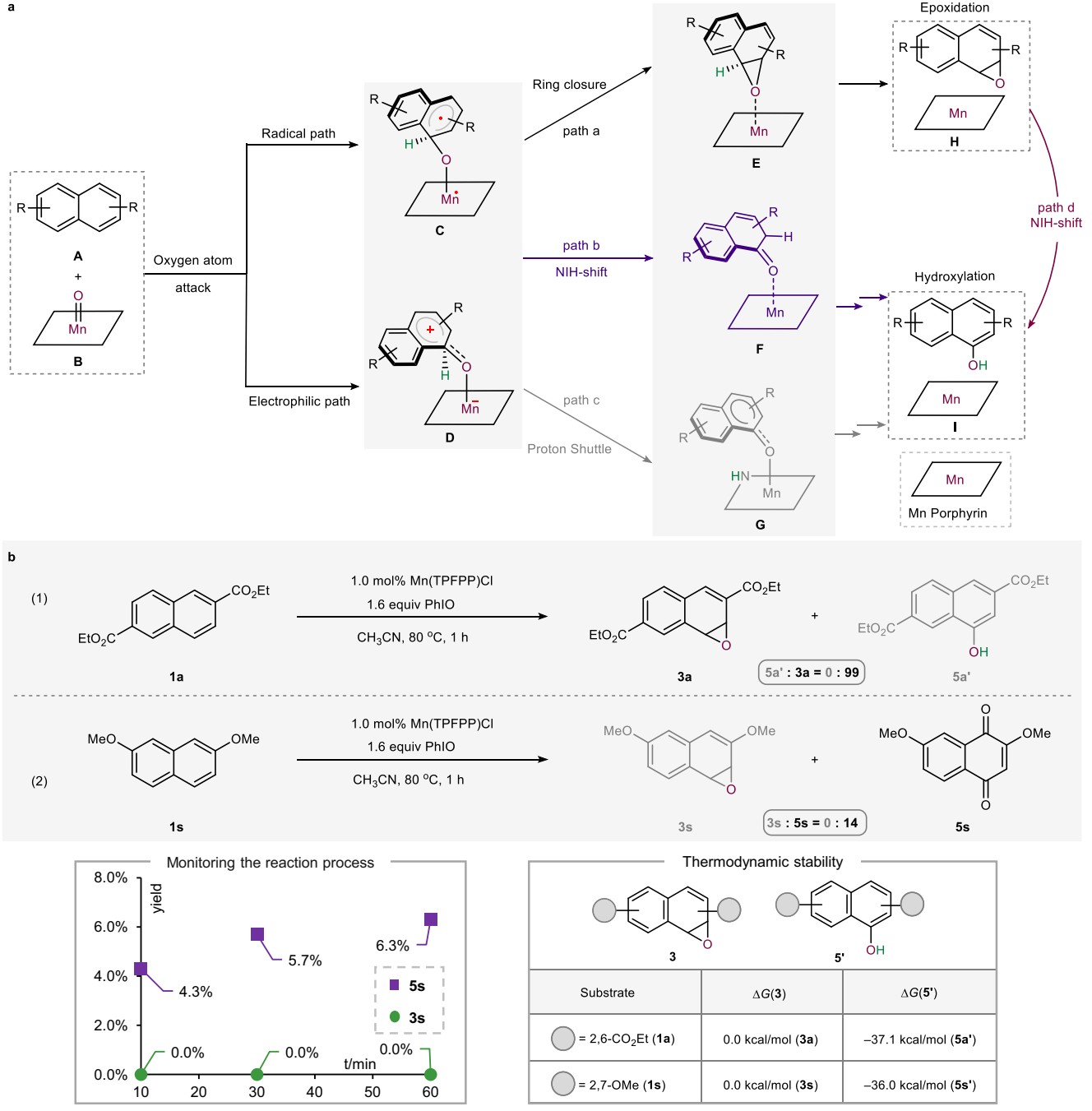

**Fig. 3 | Mechanistic consideration. a** Proposed mechanism. **b** Mechanistic experiments.

epoxides are similarly less favorable thermodynamically. Moreover, kinetic isotope effect (KIE) experiments on **1s** and [D]-**1s** indicate that C−H bond cleavage is unlikely to be the rate-determining step (see Fig. S5 for details).

Next, reaction (1), which incorporates diethyl-2,6-naphthalenedi-carboxylate **1a**, was first employed as a model to investigate the mechanism for epoxide formation. DFT-calculated free energy profile of oxidative functionalization of **1a** is given in Fig. 4. We have considered the possible spin states, including singlet (closed- and open-shell), triplet and quintet. Only the most favorable spin states are presented in Fig. 4, while other spin states and their corresponding minimum energy crossing points (MECP) are detailed in the SI (see Figs. S6−S8 for details). The reaction initiates with the coordination of the quintet Mn(TPFPP)Cl **2** with the oxidant PhIO to afford the intermediate [5]**INT1**. Subsequently, a high-valent Mn−O porphyrin

intermediate [5]**INT2**, with radical character localized on the oxygen atom, is formed via the oxidation transition state [5]**TS1** (see Fig. S9 for natural population analysis (NPA) spin population data). Thereafter, this oxygen atom initially reacts with **1a**, and then attacks the $C^4$ atom of π-system via the radical or electrophilic transition states [3]**TS2** and [3]**TS3**, leading to the radical intermediate [5]**INT3** and zwitterionic intermediate [3]**INT4**, respectively. It should be noted that for this C−O bond formation, there are four possibilities, depending on which carbon atom of arene is attacked by the oxyl group. Only the lowest-energy pathway is presented in the main text, and the other possibilities are detailed in Figs. S10, S11. The hydrogen atom abstraction (HAA)[43] between **1a** and [5]**INT2** was also considered (see Fig. S12 for details). Geometries and NPA charges of [3]**TS2** and [3]**TS3** are detailed in Fig. S13. It was found that the radical intermediate [5]**INT3** tends to undergo epoxidation through the ring closure transition state [5]**TS4**,

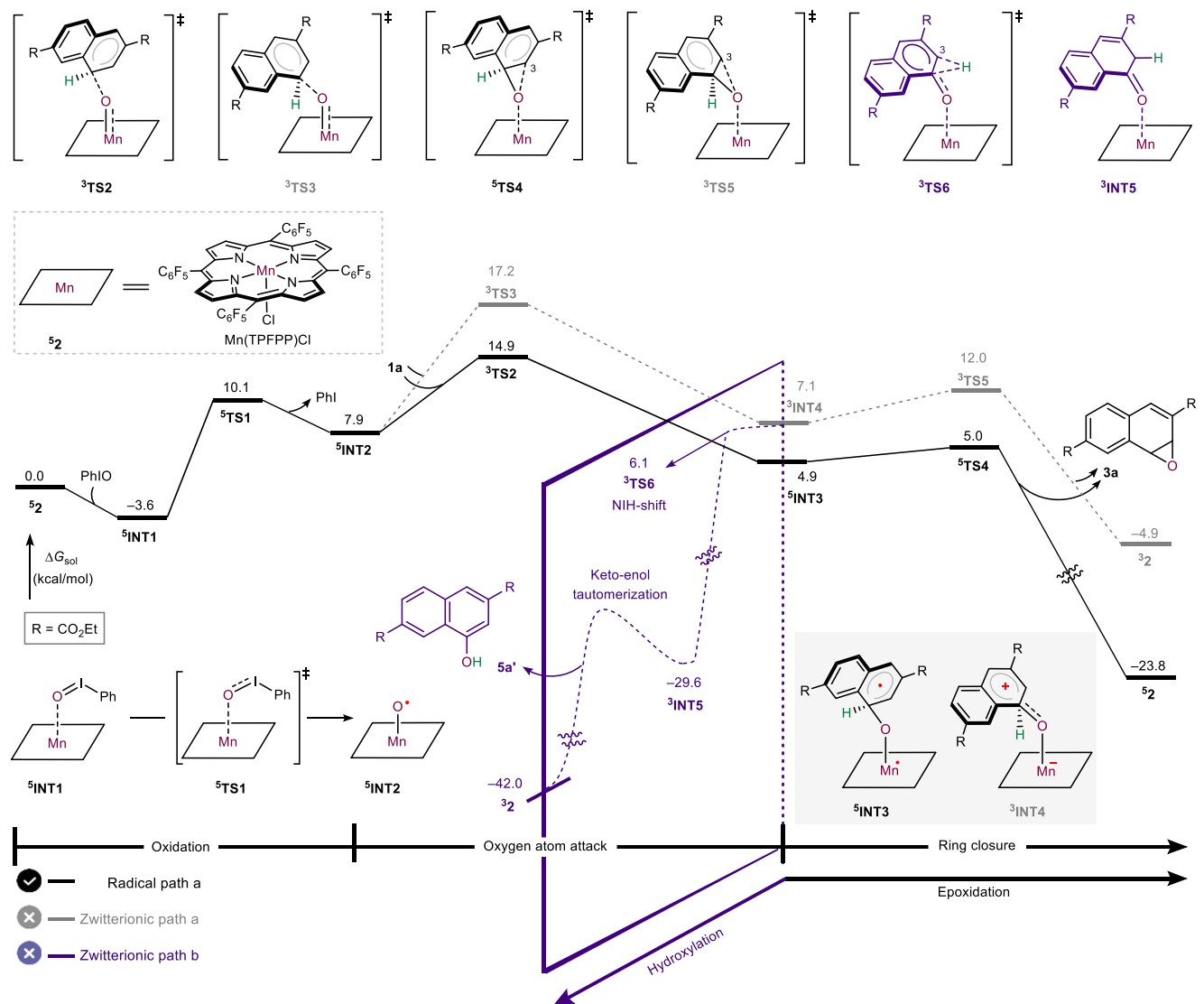

**Fig. 4 | DFT calculated free energy profile of oxidative functionalization of arene 1a.** Calculations were performed at the level of B3LYP-D3/def2-TZVP-SMD(Acetonitrile)//B3LYP-D3/6-31 G (*d*)-LANL2DZ.

delivering epoxide **3a** (Radical path a). Compared to undergoing epoxidation via the ring closure transition state **³TS5** (Zwitterionic path a), the zwitterionic intermediate **³INT4** is more inclined to experience hydroxylation through the NIH shift transition state **³TS6** (Zwitterionic path b) to form the **INT5**, which then proceeds through the keto-enol tautomerization to give the hydroxylated product **5a'**. The proton shuttle mechanism mediated by the porphyrin was calculated to be less favorable compared with **³TS6** (see the SI Fig. S14 for details). Computations indicate that the oxygen atom attack is the rate-determining step of the reaction, with the energy barrier of 18.5 kcal/mol relative to **⁵INT1**. More importantly, the energies for forming epoxide **3a** via the radical pathway through **³TS2** and **⁵TS4** are lower than those for the hydroxylated product **5a'** via the zwitterionic pathway through **³TS3** and **³TS6**, respectively. This is consistent with the chemoselectivity observed in our experiments.

To ascertain the nature of the intermediates **⁵INT3** and **³INT4**, NPA charges (Q) and spin densities (ρ) analysis were performed (Fig. 5a). Naphthalene and Mn−O porphyrin are represented in purple and green, respectively. The results reveal that **⁵INT3** exhibits single-electron distributions in the naphthalene and the Mn−O porphyrin, with the spin density of 0.918 and 3.082, respectively. It also shows minimal charge distributions for the naphthalene (Q = 0.323) and the Mn−O porphyrin

(Q = − 0.323), thereby displaying radical characteristics of **⁵INT3**. Conversely, **³INT4** has a substantial charge distribution between the naphthalene and the Mn−O porphyrin, with charges of 0.917 and −0.917, respectively, and shows an absence of single electron distribution in the naphthalene (ρ = 0.011), thus presenting zwitterionic traits. Further analysis of **⁵INT3** and **³INT4** indicate that the differences in geometries affect their distinct reactivity profiles. As shown in Fig. 5a, the $C^4 − O$ bond in **⁵INT3** is perpendicular to the plane of naphthalene ring, with an $OC^4C^{10}C^9$ dihedral angle of 89°. In the case of **³INT4**, the conjugative effect of the oxygen atom aligns the $C^4 − O$ bond in the same plane as the naphthalene, resulting in the $OC^4C^{10}C^9$ dihedral angle of 168°. This conjugation replaces the position of the hydrogen atom at the $C^4 − H$ bond with an oxygen atom of the $C^4 − O$ bond, positioning the H perpendicular to the naphthalene and placing it in an agostic state. This change facilitates the NIH shift or proton shuttle. Additionally, compared to **⁵INT3**, the $C^4 − O$ bond in **³INT4** is shortened (1.34 vs. 1.44 Å), while the Mn−O bond (1.91 vs. 1.80 Å) and $C^4 − H$ bond (1.19 vs. 1.09 Å) are elongated. These alterations bring the hydrogen atom closer to the N atom within the porphyrin and the $C^3$ atom in naphthalene, with distance of 1.94 and 1.89 Å, respectively. This further confirms the tendency of **³INT4** to undergo NIH shift and proton shuttle. In contrast, the extended $C^3 \cdots H$ (2.17 Å) and $N \cdots H$ (2.61 Å) distances in **⁵INT3** are less conducive to

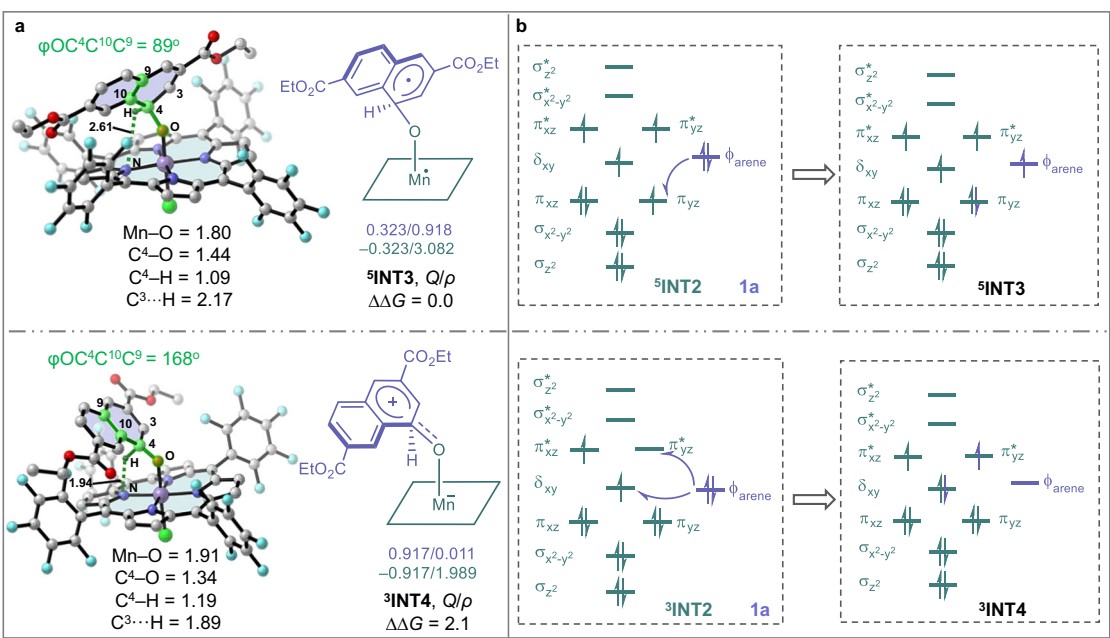

**Fig. 5 | Comparative analysis of intermediates ⁵INT3 and ³INT4. a** Geometries, NPA charges and spin densities. **b** Electron-shift diagrams for the formation of ⁵INT3 and ³INT4. Hydrogen atoms omitted for clarity. The energies and distances are shown in kcal/mol and Å, respectively.

the proton transfer, and instead, they are more likely to promote epoxidation. Moreover, the electron-shift diagrams[44–47] for the formation of ⁵INT3 and ³INT4 from arene **1a** and Mn−O porphyrin intermediate ⁵INT2 are presented in Fig. 5b. The process of forming ⁵INT3 involves transfer of a single electron from the ϕ orbital of **1a** ($\phi_{arene1a}$) to the $\pi_{yz}$ orbital in ⁵INT2. In contrast, the formation of ³INT4 requires simultaneous or sequential transfer of two electrons from the $\phi_{arene1a}$ to the $\delta_{xy}$ and $\pi^*_{yz}$ orbitals of ⁵INT2, respectively. An empty orbital on the naphthalene gives electron-deficient cationic characteristics, which enhances the conjugation with the oxygen's lone pair and significantly alters the structural configuration of ³INT4. Notably, the cationic properties of naphthalene in ³INT4, exacerbated by the electron-withdrawing ester group, reduces its stability relative to ⁵INT3, consequently enhancing the radical pathway.

Ultimately, we employed 2,7-dimethoxynaphthalene **1s** and 2,6-dimethoxynaphthalene **1t** as model substrates to explore the origin of substituent-controlled reversal of chemoselectivity. The corresponding DFT-calculated free energy profile were presented in Fig. S15-S16. The proton shuttle mediated by the porphyrin was also considered (see Fig. S17 for details). Calculations reveal that **1s** and **1t** follow the same reaction mechanism and both behave similar to the results observed with **1a**, the hydroxylation preferentially proceeds via a zwitterionic mechanism, whereas epoxidation occurs through a radical pathway. However, a reversal in reaction selectivity was observed, with hydroxylation now becoming the predominant reaction. Notably, replacing the electron-withdrawing -CO₂Et with the electron-donating -OMe reverses the stability of the radical and zwitterionic intermediates, resulting in the zwitterionic intermediate ³INT13 being 5.3 kcal/mol more stable than the radical intermediate ⁵INT12 (Fig. 6a). This trend is consistent with the behavior observed for ⁵INT8 and ³INT9 (see Fig. S18 for details). This stability reversal can be ascribed to the electron-donating effect, which significantly enhances the stability of ³INT13 and narrows the energy gap between the ϕ orbital of **1t** ($\phi_{arene1t}$) and the $\pi^*_{yz}$ orbital of ³INT2. The increased orbital energy of the $\phi_{arene1t}$ at −5.14 eV, is 1.04 eV higher than that of the $\phi_{arene1a}$ (−6.18 eV), shrinking the energy gap from 3.13 to 2.09 eV (Fig. 6b). The smaller gap facilitates electron transfer from the $\phi_{arene1t}$ to the $\pi^*_{yz}$ orbital of ³INT2, promoting formation of ³INT13 and thereby reversing the observed chemoselectivity.

## Discussion

In this work, we have developed a Mn(III) porphyrin-catalyzed method for the selective epoxidation of electron-deficient naphthalenes that is highly substrate-dependent, enabling the dearomative synthesis of naphthalene epoxides. Leveraging the electron-withdrawing substituents was crucial for precisely controlling the selectivity reversal from hydroxylation to epoxidation. Theoretical studies show that after oxidation and oxygen atom attack, both of zwitterionic and radical intermediates can be formed. Mechanistic experiments and theoretical calculations indicate that diethyl-2,6-naphthalenedicarboxylate and 2,6-dimethoxynaphthalene may follow two distinct reaction mechanisms. Electron-withdrawing -CO₂Et group on the naphthalene ring favor the formation of the radical intermediate, thereby promoting the ring closure for epoxidation. In contrast, electron-donating -OMe group stabilize the zwitterionic intermediate, which subsequently undergoes an NIH shift followed by keto−enol tautomerization to complete hydroxylation. The reversal in selectivity is mainly ascribed to the energy gap between the ϕ orbital in the arene and the $\pi^*_{yz}$ orbital of the high-valent Mn−O porphyrin. Our studies have significantly advanced the field of Mn(III) porphyrin-catalyzed epoxidation of electron-deficient naphthalene derivatives, particularly in synthetic methodologies and mechanistic studies. Currently, our laboratory is devoted to employing multivariate linear regression (MVLR)[48] analysis to unravel quantitative structure-activity relationships governing substituent effects in epoxidation and hydroxylation, thereby facilitating the synthesis of a broader array of epoxides.

## Methods

### Synthesis of TPFPP

Under a nitrogen atmosphere and protected from light, the following reagents were added sequentially to a 2 L round-bottom flask equipped with a magnetic stir bar: dichloromethane (1.5 L), freshly distilled pyrrole (1.0 g, 15 mmol, 1.0 equiv.), and pentafluorobenzaldehyde (5.88 g, 30 mmol, 2.0 equiv.). After 15 min of reaction, boron trifluoride diethyl etherate (0.71 g, 5.0 mmol, 0.33 equiv.) was added. The mixture was stirred for 3 h, followed by the addition of 2,3-dichloro-5,6-dicyano-1,4-benzoquinone (6.81 g, 30 mmol, 2.0 equiv.) at room temperature. After an additional 3 h of stirring, triethylamine (5.0 mL) was

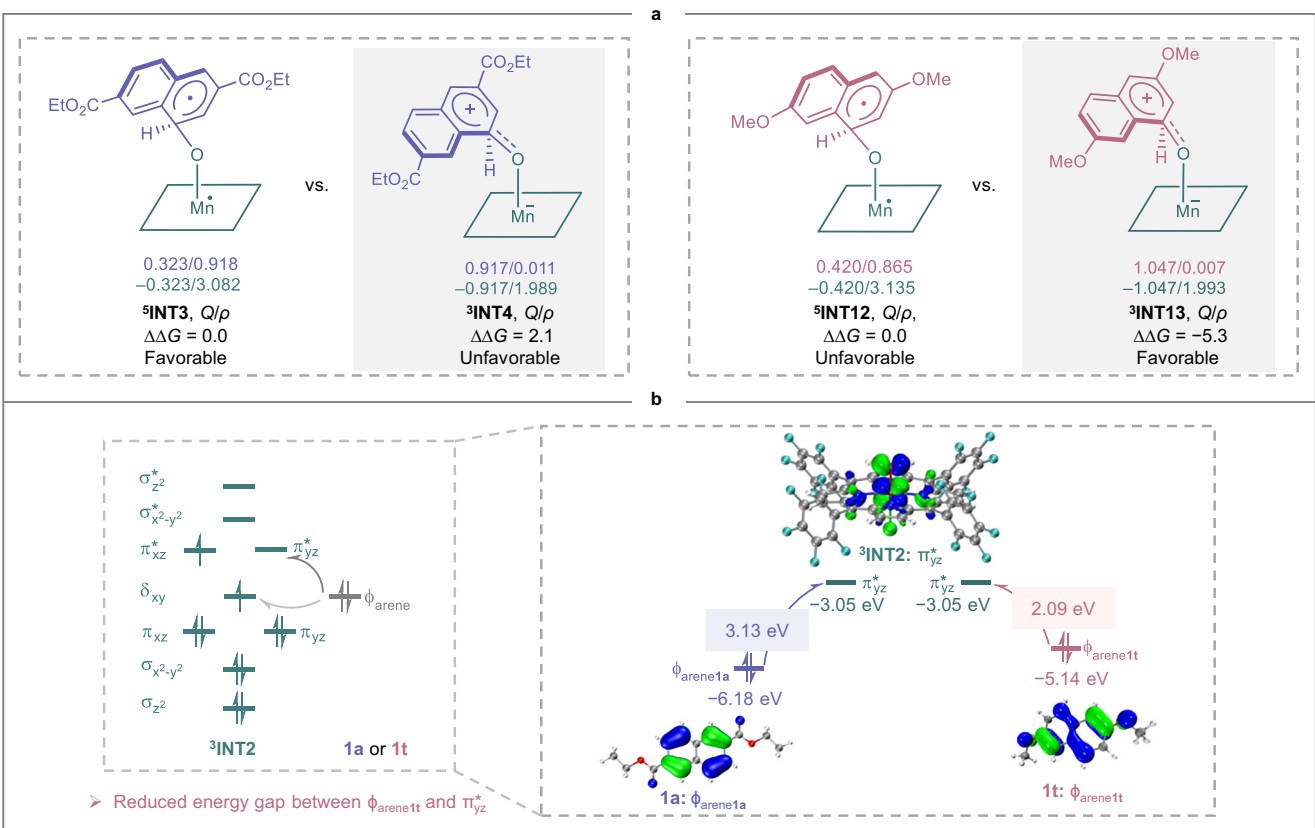

**Fig. 6 | The origin of substituent-controlled reversal of selectivity. a** Key intermediates. **b** Electron transfer from ϕ orbital of arene to $\pi^*_{yz}$ orbital of **INT2**.

added, and the mixture was stirred for another 30 minutes. The crude product was purified by silica gel column chromatography (eluent: petroleum ether/dichloromethane = 2/1, v/v) to afford TPTPP as a purple solid (443 mg, 12% yield).

### Synthesis of Mn(TPFPP)Cl

In a 100 mL round-bottom flask equipped with a stir bar and under a nitrogen atmosphere, TPFPP (974 mg, 1 mmol, 10 equiv.) and manganese(II) acetate (1.76 g, 10 mmol, 10 equiv.) were charged. Dimethylformamide (50 mL) was added to the flask, and the reaction mixture was heated at 140 °C for 24 h. After completion, the mixture was cooled to room temperature and treated with 10 mL of concentrated hydrochloric acid (36%). The resulting precipitate was collected by filtration and washed with deionized water until the filtrate became nearly colorless. The product was dried overnight under reduced pressure prior to use, affording 560 mg of a dark red solid (53% yield).

### General procedure for oxidation reaction

A 38 mL sealed tube (with a Teflon-lined cap) equipped with a magnetic stir bar was charged with substrate (1.0 mmol, 1.0 equiv.), Mn(TPFPP)Cl (0.01 mmol, 1.0 mol%), PhIO (1.6 mmol, 1.6 equiv.), and $CH_3CN$ (5.0 mL). The tube was then sealed and immersed in a preheated oil bath at 50 °C. The reaction mixture was stirred for 1 h at this temperature before being cooled to room temperature. The solvent was removed under reduced pressure to afford the crude product, which was subsequently purified by flash chromatography on silica gel using petroleum ether/ethyl acetate as the eluent to yield the target product.

### Computational methods

All DFT calculations are performed using the Gaussian16 package[49]. Geometric optimizations are performed at (U)B3LYP-D3[50–53] level of

theory. The LANL2DZ basis set[54–56] with ECP was used for Mn and I atoms, and the 6-31 G (d) basis set[57–59] was used for other atoms. Frequency analyses were also performed at the same level of theory as geometry optimization to confirm whether optimized stationary points were either local minimum or transition state, as well as to evaluate zero-point vibrational energies and thermal corrections for enthalpies and free energies at 298.15 K. At the theoretical level of (U)B3LYP-D3/def2-TZVP[60,61], the single point solvation energy is calculated using the SMD[62] solvation model (solvent = acetonitrile) on the gas phase optimized geometry. Stability tests were performed on all structures to ensure that the correct unrestricted wave functions were obtained. The CYLview was used to visualize the optimized structure[63]. Orbital analyses were performed with Multiwfn[64] and VMD[65] software package. The minimum energy crossing point (MECP) was calculated using the sobMECP program[66,67], which interfaces with Gaussian16. The NPA charges (Q) and spin densities (ρ) were computed at the (U) B3LYP-D3/6-31 G (d)-LANL2DZ level.

### Data availability

Crystallographic data for the structures reported in this Article have been deposited at the Cambridge Crystallographic Data Centre, under deposition numbers CCDC 2404381 (**3a**), 2482279 (**4e**) and 2485413 (**4j**). Copies of the data can be obtained free of charge via https://www. ccdc.cam.ac.uk/structures/. The experimental data generated in this study are provided in the Supplementary Information. The computational source data generated in this study have been deposited in the Figshare database (https://doi.org/10.6084/m9.figshare.28130495)[68]. All data are available from the corresponding author upon request.

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

## Acknowledgements
We thank the National Natural Science Foundation of China (22138011, Y.-B. She; 22371256, Y.-F. Yang), the Zhejiang Provincial Natural Science Foundation of China (LR25B020002, Y.-F. Yang), the Fundamental Research Funds for the Provincial Universities of Zhejiang (RF-C2022006, Y.-F. Yang) for financial support. We thank Jiyong Liu from Zhejiang University for the help in the single crystal measurement and analysis. We are grateful to Jiren Liu for his helpful discussion on this article.

## Author contributions
H. Wu and Y.-F. Yang analyzed the data and wrote the manuscript; J. Xu performed experiments; J. Tai performed the DFT calculations; Y. Ge and J. Gao analyzed the data; G. Li provided the experimental guidance; Y.-B. She and Y.-F. Yang conceived the idea and supervised this project. All authors discussed the results and edited the paper.

## Competing interests
The authors declare no competing interests.
