## [Transparent Peer Review file · Nature Communications]

Biomimetic Mn(III) Porphyrins-Catalyzed Aromaticity-Breaking Epoxidation of Electron-Deficient Naphthalene

Corresponding Author: Professor Yun-Fang Yang

Version 0:

Reviewer comments:

Reviewer #1

(Remarks to the Author)

This manuscript disclosed a biomimetic Mn porphyrin catalyst for the selectivity epoxidation of electron-deficient naphthalene using iodosylbenzene as an oxidant. A detailed mechanistic investigation by DFT calculations showed that the electron-withdrawing substituents play an essential role in dictating the chemoselectivity, namely epoxidation vs hydroxylation. A diradical pathway for epoxide formation was found to be preferred over the zwitterionic pathway for phenol formation. The results are of great interest and could be published after addressing the following minor issues:

1. For the first C-O bond formation, there are four possibilities, depending on which carbon atom is attacked by the oxyl group. I understand that the one presented in Figure 2 should be the most favorable one, but the other three possibilities should be calculated and presented in the supporting information.
2. What are the spin populations on O and Mn in Int2? Is it a Mn(V)=O or a Mn(IV)-O radical?
3. It would be interesting to see why oxepin (opening of the epoxide) is not formed.
4. Both triplet and quintet were considered for TS2 (diradical pathway), is it possible that the zwitterionic pathway exists in the quintet state (TS3)?

Reviewer #2

(Remarks to the Author)

The work describes the epoxidation of an electron deficient naphthalene with a manganese catalyst. The reaction is compared with the oxidation of an electron rich naphthalene, that gives exclusively a naphthol. Using DFT computations the authors propose that the epoxidation proceeds via a radical path while the hydroxylation forms via a cationic path, where a NIH shift enables recovery of the aromaticity.

The chemoselective epoxidation of a naphthalene is quite unusual because it results in functionalized dearomatized products. As well discussed by the authors, the previous work by Freire may be the single precedent besides P450's. In Freire's case the reaction forms diepoxides. The selective formation of a monoepoxide is a singular aspect of the current work. Presumably because of the EW character of the ester moieties, the removal of the aromaticity does not make the non-aromatic double bond more reactive. This makes the reaction quite interesting.

The epoxidation reaction is detailed for a single substrate and the authors focus in providing a mechanistic interpretation for the unusual chemoselectivity of the reaction. However, the mechanistic proposal almost purely relies in DFT computations. Calculations with DFT methods for red-ox reactions of first row transition metals having different spin states are subjected to uncertainties, and additional experimental evidence that could help validate the calculations is missing.

For example, one wonders if the two mechanisms may result in different KIE's and if so, if these can be experimentally corroborated. Of if different substituents and they substitution pattern (see below) behave in one way or another.

But probably the most important question that is not addressed is the relative reactivity of the epoxides. The authors assume that the final product pattern originates from different oxygen atom transfer mechanisms, and then try to explain this hypothesis by calculations. However, these may also reflect different stability of the epoxides. Authors need to demonstrate that in the case of the methoxide substituted naphthalene the formation of naphthol does not originate from a fast rearrangement of an initially formed epoxide. The comparison of the data from entries 5 and 6 points towards this possibility. When the reaction is performed during 4 h, the reaction basically gives epoxide, but as it is left for 8h, the epoxide goes down while the naphthol increases. This seems to indicate that the epoxide is converted over time in naphthol, even for the case of the EW substituted naphthalene. It is likely that the lack of epoxide detection reflects lack of stability of the epoxide. In addition, the comparison between the two substrates does not take into account the fact that the substituent substitution

pattern is different in both substrates. Again, this can be affecting stability. The effect of substitution pattern in the chemoselectivity of the reaction is a possible informative aspect that the authors could use to validate their hypothesis. The data in Table 1 detailing the optimization of the reaction requires further analyses. Authors detail a number of reactions where products 3 and 4 does not account for the substrate conversion. See for example entries 5, 6, 10 and 11. That means that other products are formed. No explanation is given. The work details quite an interesting and probably synthetically important reaction, however the mechanistic study is far from conclusive to recommend publication at this stage.

Reviewer #3

(Remarks to the Author)

The authors describe an interesting mono-epoxidation of naphthalenes through the action of an Mn(porphyrin) catalyst and iodosylbenzene as terminal oxidant. The divergent reactivity of two naphthalenes was explored – in the case of diethyl-2,6-naphthalene dicarboxylate, mono-epoxidation to yield an arene-oxide was observed, and a naphthol product was observed in the case of 2,7-dimethoxynaphthalene. Extensive DFT computations were employed to rationalize the mechanistic divergence of these two substrates. As described by the authors, the direct oxidation of aromatic systems to produce arene-oxides is well-described in the literature. However, these are highly reactive intermediates that tend to undergo further transformations – such as the NIH-shift observed by the authors. As a result, the described reaction is interesting, yet ultimately, I do not believe it constitutes a significant enough advance to warrant publication in Nature Communications. For instance, the electronic effect on the rate of the NIH-shift is well-documented (and well-cited by the authors) and can be quickly rationalized through the stabilization of positive charge in the transition state of this reaction. Therefore, at least to me, it would be readily apparent that the di-methoxy analog would be much less stable and exhibit a higher propensity to undergo the NIH-shift to yield the corresponding naphthol. Nevertheless, this work is unique in that it sets the framework for the development of methodology for the synthesis of 1,2-naphthalene-oxides. I am not convinced that two substrates are enough to fully explore this chemistry, for instance, I am curious as to what happens in the case of a mono-substituted naphthalene-carboxylate or even simple naphthalene. The diethyl-2,6-naphthalene dicarboxylate seems perhaps to be a highly tailored substrate that exhibits the desired reactivity, however it would require more convincing that this reaction could be leveraged for the preparation electron-deficient arene-oxides – which is a direct goal that the authors would like to accomplish (line 51-53). As a result, I believe the authors would have to demonstrate a reasonably-sized substrate scope of naphthalenes – or other polyaromatics – that can yield arene-oxides, as well as derivitization to prepare more functional dearomatized products.

More specific comments and questions:

- 1: Figure 1 currently reads “challenge for epoxidation:” – there are also several typos and formatting inconsistencies throughout the figures that can be polished.
- 2: Figure 2: for b) the authors mention a 99:0 selectivity and a 0:15 selectivity for naphthol – perhaps 0:99 is better and less confusing?
- 3: Currently, the results from a different paper are being used in the case of 2,7-dimethoxynaphthalene for conditions to prepare the naphthol, however the reaction conditions are pretty different from this paper. I think the authors should run that substrate under identical conditions to optimized conditions for diethyl-2,6-naphthalene dicarboxylate to make a more direct comparison. Currently, there is a 23 hour increase in reaction time, a doubling of the catalyst loading, and a very slight increase in oxidant loading – I am curious if the authors can observe the arene-oxide under optimized conditions for the arene-oxide. For example, table S3 entry 7 shows that after 24h, the primary product is naphthol, albeit at a higher temperature than optimized conditions.
- 4: The SI is suitable for publication.

Version 1:

Reviewer comments:

Reviewer #1

(Remarks to the Author)

The manuscript has been revised properly according to all reviewers' comments, therefore, it could be published as it is.

Reviewer #2

(Remarks to the Author)

The authors have thoughtfully addressed all the points I raised in my original review;

- One can argue that the lack of detection of the epoxide in the electron rich system may just reflect a faster NIH shift decomposition. However, the computed stability of the epoxides and naphthols for the dimethoxy and diester substrate is quite convincing in establishing a common higher relative stability of the naphthols.
- The difference between the dimethoxy and the diester is also convincingly demonstrated not to depend on the substitution pattern.
- My suggestion of exploring KIE's was not intended to discard hydrogen abstraction as a possible mechanism but instead, for providing experimental data that could be correlated with the computation. It is a bit surprising that KIE's >1 are measured

since the rds of the reaction involves a change from sp² to sp³ in the attacked carbon. In any case, I doubt there is data in literature to compare.

- The authors can not provide explanation for the loss of mass balance but the explanation and data provided is convincing demonstrating that multiple products are formed in small amounts.

On the other hand, the revision has brought important data that needs reconsideration in some of the claims of the authors;

- The substrate scope provided is quite nice but it also demonstrates that the monoepoxidation reaction is singular for very specific substitution patterns, besides the initially discussed electronics of the substituents. That means that the novelty of the current reaction, with respect to Freire's has been the identification of 2,6 and 2,7 diester substituted naphthalenes as substrates that provide selectively monoepoxidation. Other substituents and other substitution patterns led to phenols or proceed to diepoxidation. This is not a system that selectively delivers monoepoxides. It is an important question. Indeed, It very much compromises the significance of the current work. The use of simple naphthalene (as suggested by one reviewer) is missing.

Accordingly, authors need then to reconsider the manuscript in different places where the monoepoxidation is highlighted. For example, in the introduction section, when comparing with previous naphthalene epoxidation systems. Currently, it seems that the current system (oxidant) is the only one that provides chemoselectively monoepoxides, wo noticing that this is substrate (specially substitution pattern) dependent.

The discussion of the substrate scope also needs taking into account this discussion on the factors that define chemoselectivity (mono versus diepoxide)

The case of the sulfonate substrates 1k and 1l is quite intriguing and different from the rest of the mono substituted substrates. Monoepoxidation at takes place at the substituted ring, while diepoxidation takes place at the non substituted ring. Some coment is pertinent.

- The diepoxides can exist as two diastereoisomers. The spectroscopic data show that indeed the authors isolate a single diastereoisomer. The diastereoisomer needs to be identified and discussed in the manuscript. Graphics need to be corrected accordingly.

- When this work was being revised a manuscript has appeared that performs the enantioselective diepoxidation employing a manganese catalyst (doi.org/10.1002/anie.202504356). The connection is quite obvious and I understand that to some extend compromises the novelty and significance of the current work. This has been neglected. Authors need to make explicit mention and comparison.

- Fig 2 needs to be reconsidered. In the text the authors refer to paths and intermediates that are not shown in the figure. Paths and mentioned intermediates should be shown.

Reviewer #3

(Remarks to the Author)

The authors describe an epoxidation of naphthalenes using an Mn(porphyrin) catalyst and iodosylbenzene as terminal re-oxidant, where both mono and bisepoxidation are obtained in a highly substrate dependent manner. My initial comments were addressed, which primarily focused on the investigation of substrate scope to explore the synthetic utility of this transformation. The authors have included a substrate scope, where mono-epoxidation, bis-epoxidation, and isomerization are possible. While the distribution of products appears to be highly substrate-dependent, this is to be expected from a complex reaction of this type. Coupled with the unique computational study, I believe this work is now suitable for publication after addressing the following comments:

- 1) I do not believe the term "yield" can be applied based on total reaction conversion and the current notation is confusing. Would it be possible to report the isolated yield since all compounds were isolated?
- 2) Can the authors comment on which diastereomer is obtained (syn or anti) in the case of bis-epoxidation? Is a single diastereomer observed from the crude reaction in all cases? Since the SI indicates that single diastereomers are obtained in all cases, this should be annotated appropriately in all figures.
- 3) I do not believe Table 2 is a table, but a figure?

Version 2:

Reviewer comments:

Reviewer #3

(Remarks to the Author)

The authors addressed my previous round of inquiries through the addition of isolated yields to Figure 2 and a discussion of the diastereoselectivity of bis-epoxidation. A crystal structure was also obtained for further confirmation. All previous comments have been addressed, and I have no further inquiries. I believe the work presents an meaningful advance in the field of arene-oxidation and is suitable for publication in Nature Communications.

Reviewer #1 (Remarks to the Author)

This manuscript disclosed a biomimetic Mn porphyrin catalyst for the selectivity epoxidation of electron-deficient naphthalene using iodosylbenzene as an oxidant. A detailed mechanistic investigation by DFT calculations showed that the electron-withdrawing substituents play an essential role in dictating the chemoselectivity, namely epoxidation vs hydroxylation. A diradical pathway for epoxide formation was found to be preferred over the zwitterionic pathway for phenol formation. The results are of great interest and could be published after addressing the following minor issues:

1. For the first C-O bond formation, there are four possibilities, depending on which carbon atom is attacked by the oxyl group. I understand that the one presented in Figure 2 should be the most favorable one, but the other three possibilities should be calculated and presented in the supporting information.

Response: We sincerely thank the reviewer for this valuable suggestion. We have calculated the three alternative pathways for C–O bond formation and presented them in Fig. S10-S11 of the Supporting Information.

For the C–O bond formation in the radical process, we calculated the three other possible oxygen atom attack transition states, ${}^3\text{TS2-C}^1$ (19.7 kcal/mol), ${}^3\text{TS2-C}^2$ (25.0 kcal/mol), and ${}^3\text{TS2-C}^3$ (17.6 kcal/mol). All are significantly higher in energy than ${}^3\text{TS2}$ (14.9 kcal/mol).

Fig. S10. The possible radical attack pathways.

For the electrophilic pathway, we also explored transition states for oxygen atom attack at C^1 , C^2 , and C^3 . Despite extensive efforts, these transition states could not be located. Analysis showed that the intermediates formed by oxygen atom attack at the C^1 , C^2 , C^3 , and C^4 sites of the substrate each exhibit five resonance structures. Notably, the C^1/C^4 intermediates feature two stable

resonance forms, while C² and C³ support only one. Moreover, steric hindrance from the ortho substituent disfavors C¹ substitution, making C⁴ the optimal attack site. This suggests that the undetected transition states for the C¹, C², and C³ sites likely result from their high energy barriers.

The manuscript has been revised as follows: It should be noted that for this C–O bond formation, there are four possibilities, depending on which carbon atom of arene is attacked by the oxyl group. Only the lowest-energy pathway is presented in the main text, and the other possibilities are detailed in Fig. S10-S11.

Fig. S11. Resonance structures of electrophilic oxygen atom attack at C¹, C², C³, and C⁴ reaction site.

2. What are the spin populations on O and Mn in Int2? Is it a Mn(V)=O or a Mn(IV)-O radical?

Response: Thank you for raising this very insightful question. The natural population analysis (NPA) reveals spin populations of 0.819 on O and 2.235 on Mn in ⁵INT2, which indicates that the species is best described as a Mn(IV)-O radical. We apologize for not fully considering this initially. The structure of ⁵INT2 has been corrected in Fig. 3 of the manuscript, which helps us better understand the electronic structure of high-valent metal-oxygen intermediates.

Fig. S9. NPA spin population of ⁵INT2.

The manuscript has been revised as follows: Subsequently, a high-valent Mn-O porphyrin intermediate ⁵INT2, with radical character localized on the oxygen atom, is formed via oxidation transition state ⁵TS1 (see Fig. S9 for NPA spin population data).

3. It would be interesting to see why oxepin (opening of the epoxide) is not formed.

Response: Thank you for your question. The formation of oxepin from naphthalene oxide is disfavored due to both structural and energetic factors. This isomerization involves converting a stable aromatic system into a strained, anti-aromatic fused bicyclic structure. Our computational results show that this process is endergonic by 33.6 kcal/mol, indicating that the oxepin formation is highly thermodynamically unfavorable in our system.

The thermodynamics for the formation of oxepins

4. Both triplet and quintet were considered for TS2 (diradical pathway), is it possible that the zwitterionic pathway exists in the quintet state (TS3)?

Response: Thank you for raising this very insightful question. The zwitterionic pathway is unlikely to occur in the quintet state (TS3) within our system. In high-spin states, the frontier molecular orbitals are predominantly singly occupied, which disfavors the formation of ion-pair structures. Furthermore, our attempts to locate the high-spin zwitterionic transition state ⁵TS3 led instead converged to the quintet radical transition state ⁵TS2'.

Reviewer #2 (Remarks to the Author):

The work describes the epoxidation of an electron deficient naphthalene with a manganese catalyst. The reaction is compared with the oxidation of an electron rich naphthalene, that gives exclusively a naphthol. Using DFT computations the authors propose that the epoxidation proceeds via a radical path while the hydroxylation forms via a cationic path, where a NIH shift enables recovery of the aromaticity. The chemoselective epoxidation of a naphthalene is quite unusual because it results in functionalized dearomatized products. As well discussed by the authors, the previous work by Freire may be the single precedent besides P450s. In Freires case the reaction forms diepoxides. The selective formation of a monoepoxide is a singular aspect of the current work. Presumably because of the EW character of the ester moieties, the removal of the aromaticity does not make the non-aromatic double bond more reactive. This makes the reaction quite interesting.

1. The epoxidation reaction is detailed for a single substrate and the authors focus in providing a mechanistic interpretation for the unusual chemoselectivity of the reaction. However, the mechanistic proposal almost purely relies in DFT computations. Calculations with DFT methods for red-ox reactions of first row transition metals having different spin states are subjected to uncertainties, and additional experimental evidence that could help validate the calculations is missing. For example, one wonders if the two mechanisms may result in different KIE's and if so, if these can be experimentally corroborated. Of if different substituents and they substitution pattern (see below) behave in one way or another. But probably the most important question that is not addressed is the relative reactivity of the epoxides. The authors assume that the final product pattern originates from different oxygen atom transfer mechanisms, and then try to explain this hypothesis by calculations. However, these may also reflect different stability of the epoxides. Authors need to demonstrate that in the case of the methoxide substituted naphthalene the formation of naphthol does not originate from a fast rearrangement of an initially formed epoxide. The comparison of the data from entries 5 and 6 points towards this possibility. When the reaction is performed during 4 h, the reaction basically gives epoxide, but as it is left for 8h, the epoxide goes down while the naphthol increases. This seems to indicate that the epoxide is converted over time in naphthol, even for the case of the EW substituted naphthalene. It is likely that the lack of epoxide detection reflects lack of stability of the epoxide.

Response: We fully agree with the reviewer that DFT calculations of redox reactions for first-row transition metals with different spin states are subjected to uncertainty, and that experimental validation is essential. Following your suggestions, we conducted additional mechanistic experiments to support our computational findings. First, to investigate the potential involvement of an epoxide intermediate, we monitored the reaction of 2,7-dimethoxynaphthalene (1s) over time. Thin-layer chromatography and mass spectrometry showed that quinone product 5s formed within 1–10 minutes, no detectable intermediate spots between 1s and 5s including the epoxide 3s (Fig. S4a). After extended

monitoring (10–60 minutes), no trace of 3s was detected. While this does not definitively exclude the transient formation of an epoxide, the data suggest it is not an intermediate in this case. Additionally, we calculated the relative thermodynamic stability of the epoxide 3 and hydroxylation product 5'. For substrate 1a and 1s, the hydroxylation products 5a' and 5s' were more stable than the corresponding epoxides 3a and 3s by 37.1 and 36.0 kcal/mol, respectively. This suggests that both electron-rich and electron-poor epoxides are similarly less favorable thermodynamically. Importantly, our DFT results suggest that these substrates follow distinct reaction mechanisms: 1a reacts via a radical pathway leading epoxidation, while 1s predominantly undergoes hydroxylation through a zwitterionic pathway. Taken together, these findings demonstrate that the formation of the hydroxylation product may bypass an epoxide intermediate for 1s. **The updated results have been incorporated into the manuscript (Fig. 2) and the Supporting Information (Fig. S4).**

Fig. S4 a) Monitoring the reaction process of 1s. petroleum ether/ethylacetate = PE/EA. The yields were determined by ¹H NMR analyses of the crude products using CH₂Br₂ as the internal standard. b) the thermodynamic stability of epoxide 3 and hydroxylation product 5'. c) DFT calculations.

Fig. 2 | Proposed mechanism and the mechanistic experiments.

To directly address the stability of epoxide intermediates, we examined the behavior of isolated 3a, derived from 1a, under catalytic and ambient conditions. Under catalytic conditions, 3a gradually rearranged to the corresponding hydroxylated product. However, when stored at room temperature without catalyst for 34 days, only 14.4% conversion was observed (Fig. S1), suggesting that the epoxide is inherently stable and that rearrangement is catalytically driven. From this conversion, the hydroxylation yield was calculated to be 66.7%.

Fig. S1 The reaction of 3a in the absence of catalyst.

We also conducted kinetic isotope effect (KIE) experiments on 1s and [D]-1s (Fig. S5). The observed KIE value of 1.025 indicates that C–H bond cleavage is unlikely to be the rate-determining step, which is consistent with our computational results. Unfortunately, a deuterated analog of 1a was not available, so KIE experiments were limited to 1s. Finally, regarding substituent effects:

while different substitution patterns do influence reactivity, 2,6- and 2,7-dimethoxynaphthalenes show similar behavior, which we address in detail in the next response.

Fig. S5 The kinetic isotope effect (KIE) experiments of 1s and [D]-1s. The yields were determined by ^1H NMR analyses of the crude products using CH_2Br_2 as the internal standard.

2. In addition, the comparison between the two substrates does not take into account the fact that the substituent substitution pattern is different in both substrates. Again, this can be affecting stability. The effect of substitution pattern in the chemoselectivity of the reaction is a possible informative aspect that the authors could use to validate their hypothesis.

Response: Thanks for your comments. To evaluate the influence of substitution pattern on chemoselectivity, we conducted comparative experiments using 2,7-dimethoxynaphthalene 1s and 2,6-dimethoxynaphthalene 1t under identical reaction conditions (Fig. S3a). Both substrates afforded the corresponding quinone products in comparable yields. Moreover, we performed DFT calculations on the relative stability of the epoxide and hydroxylation product derived from each substrate (Fig. S3b). The results showed that both substitution patterns lead to products with comparable stabilities. Taken together, these experimental and computational results indicate that the observed chemoselectivity is not strongly affected by the substitution pattern. **The relevant data have been included in the supporting information.**

Fig. S3. The effect of substitution pattern.

4. The data in Table 1 detailing the optimization of the reaction requires further analyses. Authors detail a number of reactions where products 3 and 4 does not account for the substrate conversion. See for example entries 5, 6, 10 and 11. That means that other products are formed. No explanation is given.

Response: We appreciate the reviewer's observation. As correctly noted, in several entries of Table 1 (e.g., 5, 6, 10, and 11), the yields of products 3 and 4 do not fully account for the substrate conversion, suggesting the formation of additional, uncharacterized by-products. Arene oxidation is inherently complex and often generates a variety of side products beyond the reported epoxides and phenols—including dihydroxylated compounds, diepoxides, and quinones. We made efforts to isolate and identify these possible by-products. However, due to the complexity of the reaction mixtures, the presence of isomeric species, and their closely related polarities, effective separation and characterization were not achievable under our current conditions. We believe that these uncharacterized by-products are responsible for the observed mass imbalance. This point will be explicitly noted in the revised manuscript in the discussion of Table 1.

5. The work details quite an interesting and probably synthetically important reaction, however the mechanistic study is far from conclusive to recommend publication at this stage.

Response: We appreciate your insightful comments and acknowledge that our initial findings were indeed at a preliminary stage. To establish significance and strengthen conclusions, comprehensive additional experiments were conducted, including substrate expansion, mechanistic experiments, and DFT studies. The effective integration of experiments and theoretical calculations provides mutual validation, enhancing overall research credibility. We sincerely hope these revisions address all concerns and the work will merit publication.

Reviewer #3 (Remarks to the Author)

The authors describe an interesting mono-epoxidation of naphthalenes through the action of an Mn(porphyrin) catalyst and iododibenzene as terminal oxidant. The divergent reactivity of two naphthalenes was explored – in the case of diethyl-2,6-naphthalene dicarboxylate, mono-epoxidation to yield an arene-oxide was observed, and a naphthol product was observed in the case of 2,7-dimethoxynaphthalene. Extensive DFT computations were employed to rationalize the mechanistic divergence of these two substrates. As described by the authors, the direct oxidation of aromatic systems to produce arene-oxides is well-described in the literature. However, these are highly reactive intermediates that tend to undergo further transformations – such as the NIH-shift observed by the authors. As a result, the described reaction is interesting, yet ultimately, I do not believe it constitutes a significant enough advance to warrant publication in Nature Communications. For instance, the electronic effect on the rate of the NIH-shift is well-documented (and well-cited by the authors) and can be quickly rationalized through the stabilization of positive charge in the transition state of this reaction. Therefore, at least to me, it would be readily apparent that the di-methoxy analog would be much less stable and exhibit a higher propensity to undergo the NIH-shift to yield the corresponding naphthol. Nevertheless, this work is unique in that it sets the framework for the development of methodology for the synthesis of 1,2-naphthalene-oxides. I am not convinced that two substrates are enough to fully explore this chemistry, for instance, I am curious as to what happens in the case of a mono-substituted naphthalene-carboxylate or even simple naphthalene. The diethyl-2,6-naphthalene dicarboxylate seems perhaps to be a highly tailored substrate that exhibits the desired reactivity, however it would require more convincing that this reaction could be leveraged for the preparation electron-deficient arene-oxides – which is a direct goal that the authors would like to accomplish (line 51-53). As a result, I believe the authors would have to demonstrate a reasonably-sized substrate scope of naphthalenes – or other polyaromatics – that can yield arene-oxides, as well as derivitization to prepare more functional dearomatized products.

Response: We thank the reviewer for your insightful comments. We agree that electron-rich arenes, such as 2,7-dimethoxynaphthalene, are more prone to undergo the NIH-shift due to greater stabilization of the developing positive charge in the transition state. While the NIH-shift mechanism is well-established in enzymatic reactions, it remains underexplored in biomimetic catalysis. Moreover, achieving selective arene epoxidation persists as a significant challenge, particularly for electron-deficient naphthalenes. We appreciate the reviewer's perspective that the two substrates presented are insufficient to comprehensively explore this chemistry, and a broader substrate scope is required. Accordingly, we have conducted extensive additional experiments, including substrate expansion, mechanistic studies, and DFT calculations to thoroughly investigate this chemistry. These new results are now incorporated in the revised manuscript and Supporting Information.

With optimal conditions established, we systematically investigated the reactivity of various naphthalene derivatives, focusing on both substitution pattern and electronic effects (Table 2). Our prior work (Tetrahedron Lett. 2023, 123, 154535) had already indicated that electron-rich naphthalenes preferentially yield quinone products under similar conditions. Consistent with this, when simple naphthalene was subjected to the current reaction conditions, it mainly produced quinones (Fig. S2), indicating that electron-donating groups disfavor epoxide formation. We therefore turned our attention to electron-deficient substrates (Table 2).

Fig. S2 The reaction of naphthalene.

Table 2 | Electron-deficient arenes scope. The yields were based on the conversion yields.

Among di-substituted naphthalenes, 2,6- and 2,7-diester-substituted derivatives afforded mono-epoxides 3a, 3b and 3c in moderate yields. In contrast, methyl 6-bromo-2-naphthoate afforded 3d in low yield. Cyano-substituted derivatives such as 2,3- and 1,4-cyano-substituted naphthalenes predominantly formed diepoxides (4e and 4f), while methyl 5-bromo-2-naphthoate yielded a mixture of epoxide 3g and quinone 5g. For mono-substituted naphthalenes, only sulfonate-substituted substrates (-SO₃Ph or -SO₃Me derivatives) produced mono-epoxides (3k, 3l), along with diepoxides (4k, 4l). Other electron-withdrawing groups such as -CO₂Me, -CN, and NO₂ primarily gave diepoxides or mixtures of diepoxides and quinones (4h/5h, 4i/5i, 4j/5j, 4m). Mono-substituted quinoline derivatives bearing -CO₂Me or -CN mainly afforded dioxetanes 4n and 4o. Notably, no epoxides were detected for naphthalene-2,6-dicarbonitrile 1p, 2-(trifluoromethyl) naphthalene 1q, or 1-(trifluoromethyl) naphthalene 1r.

More specific comments and questions:

1: Figure 1 currently reads “challgegs for epoxidation:” – there are also several typos and formatting inconsistencies throughout the figures that can be polished.

Response: Thank you for pointing out the typos and the formatting inconsistencies. We have carefully examined and refined Figure 1, correcting "challgegs" to "challenges", "chemoselectivie" to "chemoselective", "mild conditions" to "mild condition", "dearomative" to "involving dearomatization", "chemoselectivie" to "high chemoselectivity".

Fig. 1 | The development of oxidative functionalization of arenes.

2: Figure 2: for b) the authors mention a 99:0 selectivity and a 0:15 selectivity for naphthol – perhaps 0:99 is better and less confusing?

Response: Thank you for the helpful suggestion. We agree that presenting the selectivity as 0:99 is clearer and avoids confusion. We have updated Figure 2b accordingly in the revised manuscript.

Fig. 2 | Proposed mechanism and the mechanistic experiments.

3: Currently, the results from a different paper are being used in the case of 2,7-dimethoxynaphthalene for conditions to prepare the naphthol, however the reaction conditions are pretty different from this paper. I think the authors should run that substrate under identical conditions to optimized conditions for diethyl-2,6-naphthalene dicarboxylate to make a more direct comparison. Currently, there is a 23 hour increase in reaction time, a doubling of the catalyst loading, and a very slight increase in oxidant loading – I am curious if the authors can observe the arene-oxide under optimized conditions for the arene-oxide. For example, table S3 entry 7 shows that after 24 h, the primary product is naphthol, albeit at a higher temperature than optimized conditions.

Response: Thank you for this valuable suggestion. To allow a direct comparison, we subjected 2,7-dimethoxynaphthalene 1s to the same reaction conditions optimized for diethyl-2,6-naphthalene dicarboxylate. The experiment showed that arene-oxide was not observed. In fact, time-course monitoring at 1–10 minutes and 10–60 minutes confirmed the absence of any epoxide formation under this reaction condition.

b) Monitoring the reaction process

4: The SI is suitable for publication.

Response: We appreciate the positive feedback. All suggestions have been implemented through comprehensive additional experiments, including substrate scope expansion, mechanistic investigations, and DFT studies to thoroughly investigate this chemistry. These revisions significantly strengthen the manuscript's scientific rigor and reliability. We sincerely believe that they address the reviewers' concerns and hope this work merits publication.

Reviewer #1 (Remarks to the Author):

The manuscript has been revised properly according to all reviewers' comments, therefore, it could be published as it is.

Response: We sincerely thank the reviewer for your positive feedback and recognition.

Reviewer #2 (Remarks to the Author):

The authors have thoughtfully addressed all the points I raised in my original review;
-One can argue that the lack of detection of the epoxide in the electron rich system may just reflect a faster NIH shift decomposition. However, the computed stability of the epoxides and naphthols for the dimethoxy and diester substrate is quite convincing in establishing a common higher relative stability of the naphthols.

Response: We are truly honored to receive your positive feedback.

-The difference between the dimethoxy and the diester is also convincingly demonstrated not to depend on the substitution pattern.

Response: We sincerely thank the reviewer for your recognition.

-My suggestion of exploring KIE's was not intended to discard hydrogen abstraction as a possible mechanism but instead, for providing experimental data that could be correlated with the computation. It is a bit surprising that KIE's >1 are measured since the rds of the reaction involves a change from sp² to sp³ in the attacked carbon. In any case, I doubt there is data in literature to compare.

Response: We greatly appreciate the reviewer's feedback. Indeed, we aimed to employ KIE experiments to correlate with computational results. Although there is a lack of directly comparable systems in the literature, we postulate that the observed KIE > 1 suggests a complex, multi-step process involving significant C-H bond cleavage in the transition state. Furthermore, potential contributions from secondary kinetic isotope effects or subtle experimental variations under the reaction conditions cannot be entirely ruled out.

-The authors can not provide explanation for the loss of mass balance but the explanation and data provided is convincing demonstrating that multiple products are formed in small amounts.

Response: We sincerely thank the reviewer for your positive feedback.

On the other hand, the revision has brought important data that needs reconsideration in some of the claims of the authors;

1. The substrate scope provided is quite nice but it also demonstrates that the

monoepoxidation reaction is singular for very specific substitution patterns, besides the initially discussed electronics of the substituents. That means that the novelty of the current reaction, with respect to Freire's has been the identification of 2,6 and 2,7 diester substituted naphthalenes as substrates that provide selectively monoepoxidation. Other substituents and other substitution patterns led to phenols or proceed to diepoxidation. This is not a system that selectively delivers monoepoxides. It is an important question. Indeed, it very much compromises the significance of the current work. The use of simple naphthalene (as suggested by one reviewer) is missing.

Response: We sincerely thank the reviewer for your insightful comments. We agree that the selectivity of our catalytic system is highly dependent on the substrate's specific substitution pattern, and that it does not serve as a universal system for monoepoxidation. Nevertheless, the significance of our work primarily lies in the successful isolation of several challenging electron-deficient monoepoxides under this catalytic system for the first time, complemented by detailed mechanistic studies that elucidate the origin of the selectivity control. These findings offer important mechanistic insights and design principles for the synthesis of epoxides. In the future studies, we will strengthen the integration of theoretical and experimental approaches to develop more general catalytic systems for epoxidation, thereby addressing the longstanding challenge of synthesizing arene epoxides.

In fact, during our previous revision in response to Reviewer 3, we had already investigated the simple naphthalene substrate **1u**. Accordingly, we incorporated a statement in the main text: when simple naphthalene was subjected to the current reaction conditions, quinones were obtained as the major products (see Fig. S2 for details).

Fig. S2 The reaction of naphthalene.

2. Accordingly, authors need then to reconsider the manuscript in different places where the monoepoxidation is highlighted. For example, in the introduction section, when comparing with previous naphthalene epoxidation systems. Currently, it seems that the current system (oxidant) is the only one that provides chemoselectively monoepoxides,

wo noticing that this is substrate (specially substitution pattern) dependent.

Response: Thank you for this thoughtful comment. We fully agree that it is necessary to modify our phrasing throughout the manuscript regarding the monoepoxidation by adding limiting terms that emphasize the critical role of the substrate, particularly the specific substituents and their substitution patterns. The relevant sections in the main text have been revised accordingly.

In the abstract section: “Herein, we report the first synthesis and characterization of several electron-deficient naphthalene epoxides that disrupt the aromaticity of naphthalene. Their formation is highly dependent on the substrate, especially on the specific substitution pattern.”

In the introduction section: “Encouragingly, our present work demonstrates that diethyl-2,6-naphthalenedicarboxylate, featuring a distinct substitution pattern undergoes epoxidation under mild catalytic conditions, enabling the dearomative synthesis of the corresponding epoxide with high chemoselectivity.”

In the discussion section: “In this work, we have developed a Mn(III) porphyrin-catalyzed method for the selective epoxidation of electron-deficient naphthalenes that is highly substrate-dependent, enabling the dearomative synthesis of epoxides.”

3. The discussion of the substrate scope also needs taking into account this discussion on the factors that define chemoselectivity (mono versus diepoxide)

Response: We sincerely thank the reviewer for this excellent suggestion.

Following the reviewer’s advice, we have expanded the discussion of the substrate scope (Figure 2) to explicitly address the structural factors governing chemoselectivity between mono- and diepoxidation. The revised manuscript now includes a concise rationale that directly correlates the observed selectivity with the electronic and steric characteristics of the substrates. Specifically, we have added the following discussion:

“Experimental results demonstrate that the type and substitution pattern of substituents on the naphthalene ring effectively modulate the selectivity between mono- and diepoxides formation. Substrates 1a–1d and 1g, bearing electron-withdrawing groups at the 2,5-, 2,6-, and 2,7-positions, afforded monoepoxides exclusively without diepoxidation, a behavior ascribed to the combined effects of enhanced electrophilicity and sterically constrained environment at the substituted ring. In contrast, substrates 1e, 1f, 1h–1j, and 1m–1o underwent

selective diepoxidation on the unsubstituted ring, highlighting the influence of steric accessibility. Interestingly, the sulfonate-substituted substrates **1k** and **1l** yielded mixtures of mono- and diepoxides: monoepoxidation occurred preferentially on the substituted ring, likely directed by the sulfonate group, while diepoxidation the unsubstituted ring due to steric accessibility.”

Figure 2 | Electron-deficient arenes scope. ^aThe yields were based on the conversion yields. ^bThe yields were based on isolated yields.

4. The case of the sulfonate substrates **1k** and **1l** is quite intriguing and different from the rest of the mono substituted substrates. Monoepoxidation at takes place at the substituted ring, while diepoxidation takes place at the non substituted ring. Some coment is pertinent.

Response: We sincerely thank the reviewer for this insightful comment and for highlighting the unique behavior of the sulfonate substrates **1k** and **1l**. We agree that their reactivity is distinctive, and we have provided a more detailed discussion

in Response 3 above.

5. The diepoxides can exist as two diastereoisomers. The spectroscopic data show that indeed the authors isolate a single diastereoisomer. The diastereoisomer needs to be identified and discussed in the manuscript. Graphics need to be corrected accordingly.

Response: Thank you for your insightful comments. We have carefully re-examined the stereochemical assignments of all diepoxides and confirmed that they adopt the anti-configuration based on a combination of ¹H NMR spectroscopy and single-crystal X-ray diffraction analysis. Figure 2 have been revised accordingly.

Figure 2 | Electron-deficient arenes scope. ^aThe yields were based on the conversion yields.

^bThe yields were based on isolated yields.

Based on their ¹H NMR patterns, the diepoxides fall into two distinct categories. The first group (compounds 4h–4o) exhibits coupling constants of ~2.0 Hz, consistent with the anti-configuration (see *Angew. Chem. Int. Ed.*, 1976, 88,

268). In contrast, the second group (4e and 4f) displays similar overlapping signals of the four vicinal epoxy protons, making it difficult to determine the syn/anti configuration by NMR alone. Single-crystal X-ray diffraction of representative compounds 4e and 4j unambiguously established their anti-configuration. Because 4e and 4f share identical ¹H NMR patterns, and 4j shows the same spectral characteristics as 4h–4i and 4k–4o, we confidently assign the anti-configuration to all diepoxides in this series.

6. When this work was being revised a manuscript has appeared that performs the enantioselective diepoxidation employing a manganese catalyst (doi.org/10.1002/anie.202504356). The connection is quite obvious and I understand that to some extent compromises the novelty and significance of the current work. This has been neglected. Authors need to make explicit mention and comparison.

Response: We sincerely thank you for bringing this highly relevant and important study (doi.org/10.1002/anie.202504356) to our attention. Following your suggestion, we have explicitly cited and discussed this work in the revised *Introduction*.

“Notably, while revising our manuscript, Costas and co-workers reported a breakthrough manganese-catalyzed enantioselective diepoxidation of a broad range of arenes.³⁶ Their success arises from the synergistic combination of an electron-rich, sterically demanding manganese complex and amino acid co-ligands containing a tert-butyl-leucine moiety. In contrast, our study employs a highly fluorinated Mn(TPFPP)Cl catalyst operating without any co-catalyst, enabling selective mono- and diepoxidation of electron-deficient naphthalenes. It is particularly intriguing that these distinct catalytic systems afford analogous epoxidation products. Moreover, our work complements this recent advance by providing detailed mechanistic insights into the competition between epoxidation and hydroxylation pathways, thereby elucidating the origin of substituent-controlled selectivity reversal.”

7. Fig 2 needs to be reconsidered. In the text the authors refer to paths and intermediates that are not shown in the figure. Paths and mentioned intermediates should be shown.

Response: We sincerely thank you for your valuable feedback. As suggested, we have now included the missing paths and key intermediates in the revised Figure 3a. Corresponding revisions have also been made in the main text as follows:

Fig. 3 | Proposed mechanism and the mechanistic experiments.

“It is proposed that both intermediates C and D can undergo a ring closure process (path a) to form the epoxide-coordinated intermediate E. Alternatively, they can undergo hydroxylation via either an "NIH shift" (path b) or a "proton shuttle" mechanism (path c). The NIH shift involves the intramolecular migration of a hydrogen atom (highlighted in green) to an adjacent carbon atom, yielding the ketone intermediate F, which subsequently undergoes keto-enol tautomerization. The proton shuttle mechanism involves a proton transfer mediated by the porphyrin ligand, whereby the proton first migrates to a nitrogen atom on the porphyrin ring to form the N-protonated intermediate G, and is then shuttled to the oxygen atom of the Mn-oxo intermediate species.”

Reviewer #3 (Remarks to the Author):

The authors describe an epoxidation of naphthalenes using an Mn(porphyrin) catalyst and iodosylbenzene as terminal re-oxidant, where both mono and bisepoxidation are

obtained in a highly substrate dependent manner. My initial comments were addressed, which primarily focused on the investigation of substrate scope to explore the synthetic utility of this transformation. The authors have included a substrate scope, where mono-epoxidation, bis-epoxidation, and isomerization are possible. While the distribution of products appears to be highly substrate-dependent, this is to be expected from a complex reaction of this type. Coupled with the unique computational study, I believe this work is now suitable for publication after addressing the following comments:

1) I do not believe the term “yield” can be applied based on total reaction conversion and the current notation is confusing. Would it be possible to report the isolated yield since all compounds were isolated?

Response: Thank you for this valuable suggestion. In the revised manuscript, we have now explicitly reported the isolated yields for all compounds in Figure 2 to prevent any further misunderstanding. Notably, we have confirmed that all diepoxides adopt the anti-configuration (see the following Response 2 for details).

Figure 2 | Electron-deficient arenes scope. ^aThe yields were based on the conversion yields.

^bThe yields were based on isolated yields.

2) Can the authors comment on which diastereomer is obtained (syn or anti) in the case of bis-epoxidation? Is a single diastereomer observed from the crude reaction in all cases? Since the SI indicates that single diastereomers are obtained in all cases, this should be annotated appropriately in all figures.

Response: We have confirmed that all diepoxides adopt the anti-configuration through a combination of ¹H NMR spectroscopy and X-ray diffraction analysis. Figure 2 have been revised accordingly.

Based on their ¹H NMR patterns, the diepoxides fall into two distinct categories. The first group (compounds 4h–4o) exhibits coupling constants of ~2.0 Hz, consistent with the anti-configuration (see *Angew. Chem. Int. Ed.*, 1976, 88, 268). In contrast, the second group (4e and 4f) displays similar overlapping signals of the four vicinal epoxy protons, making it difficult to determine the syn/anti configuration by NMR alone. Single-crystal X-ray diffraction of representative compounds 4e and 4j unambiguously established their anti-configuration. Because 4e and 4f share identical ¹H NMR patterns, and 4j shows the same spectral characteristics as 4h–4i and 4k–4o, we confidently assign the anti-configuration to all diepoxides in this series.

While we cannot definitively rule out the presence of trace amounts of the syn-diastereomer in the crude reaction mixture, only the anti-diastereomer was isolated and purified in all cases. And NMR analysis of the crude product did not show detectable signals corresponding to the syn-isomer, suggesting a single diastereomer isolated. We have now clearly annotated all relevant figures in the main text and Supporting Information to indicate that a single anti-diastereomer was obtained in all isolated products.

3) I do not believe Table 2 is a table, but a figure?

Response: We fully agree with the reviewer that Table 2 is not a table, but a figure. We have corrected Table 2 to Figure 2 in the revised manuscript.

Reviewer #3 (Remarks to the Author):

The authors addressed my previous round of inquiries through the addition of isolated yields to Figure 2 and a discussion of the diastereoselectivity of bis-epoxidation. A crystal structure was also obtained for further confirmation. All previous comments have been addressed, and I have no further inquiries. I believe the work presents a meaningful advance in the field of arene-oxidation and is suitable for publication in Nature Communications.

Response: We sincerely thank the reviewer for your positive feedback and recognition.